# Predictors of medical staff's knowledge, attitudes and behavior of dysphagia assessment: A cross-sectional study

Juanhui Chen[1]*, Wenqiu Ye[2], Xingyun Zheng[3], Wenna Wu[4], Yuebao Chen[5], Yinjuan Chen[6]

1 Department of Nursing, Longgang Central Hospital of Shenzhen, Shenzhen, Guangdong Province, China, 2 Department of Neurology, Longgang Central Hospital of Shenzhen, Shenzhen, Guangdong Province, China, 3 Department of Neurosurgery, Longgang Central Hospital of Shenzhen, Shenzhen, Guangdong Province, China, 4 Department of Rehabilitation, Longgang Central Hospital of Shenzhen, Shenzhen, Guangdong Province, China, 5 Department of Neurology, Guangxi International Zhuang Medicine Hospital, Nanning, Guangxi Province, China, 6 Department of Nursing, Shenzhen FuYong People's Hospital, Shenzhen, Guangdong Province, China

* pugongyingchenjuan@163.com

**Data Availability Statement:** The data supporting the findings of this study are available in the Dryad Digital Repository (DOI: https://doi.org/10.5061/dryad.djh9w0w70).

## Abstract

### Purpose

This study aimed to explore the predictors of medical staff's Knowledge, Attitudes and Behavior of dysphagia assessment, to provide reference suggestions for constructing the training program and improving the rate of dysphagia assessment.

### Methods

This study was a cross-sectional study. A total of 353 nurses and doctors from four provinces (Guangdong, Hunan, Guangxi, and Shaanxi) who were selected by convenience sampling and invited to complete the questionnaire through WeChat, DingTalk, and Tencent instant messenger from May 23 to 31, 2022. A self-reported questionnaire was used to assess participants' Knowledge, Attitude and Behavior regarding dysphagia assessment. Participants' sociodemographic, training, and nursing experience were measured using the general information sheet and analyzed as potential predictors of medical staff's Knowledge, Attitudes and Behavior of dysphagia assessment. A multiple linear regression model was used to identify the predictors.

### Results

The mean scores for Knowledge, Attitudes and Behavior of dysphagia assessments were (15.3±2.7), (35.9±4.9) and (41.4±14.4) respectively. Knowledge and Behavior of medical staff were medium, and attitude was positive. Multiple linear regression results indicated that experience in nursing patients with dysphagia, related training for dysphagia, working years in the field of dysphagia related diseases, specialized training (geriatric, swallowing and rehabilitation) and department (Neurology, Rehabilitation, Geriatrics) were significant predictors of Behavior, accounting for 31.5% of the variance. Working years in the field of

**Funding:** Medical and Health Technology Plan of Longgang District, Shenzhen. Funding was rewarded to Juanhui Chen. The Grant award number was LGKCYLWS2020128. URL: http://www.lg.gov.cn/bmzz/kjj/xxgk/qt/tzgg/content/post_7970194.html The funders play no role in the study design, data collection and analysis, decision to publish, or preparation of the manuscript.

**Competing interests:** The authors have declared that no competing interests exist.

**Abbreviations:** A, Attitudes; B, Behavior; K, Knowledge; MS-KAB-DA, Questionnaire of Medical Staff's Knowledge, Attitudes and Behavior of dysphagia assessment.

dysphagia related diseases, department (Neurology, Rehabilitation, Geriatrics) and title were significant predictors of medical staff's knowledge, accounting for 7.8% of variance. Education, experience in nursing patients with dysphagia, department (Neurology, Rehabilitation, Geriatrics) and related training for dysphagia were significant predictors of medical staff's attitude, accounting for 12.9% of variance.

## Conclusions

The study findings implied that nursing experience, training, and work for patients with swallowing disorders could have positive effects on the Knowledge, Attitudes and Behavior of medical staff regarding dysphagia assessment. Hospital administrators should provide relevant resources, such as videos of dysphagia assessment, training centers for the assessment of dysphagia, and swallowing specialist nurses.

## Introduction

Worldwide, there are approximately 80.1 million stroke survivors, with an estimated 13.7 million new cases of stroke every year. Stroke remained the second-leading cause of death and the third-leading cause of death and disability combined [1]. Dysphagia occurs in 30.0–78.0% of patients after stroke [2–4]. Dysphagia may be defined as the partial or complete inability to prepare and move a bolus of food, fluids, or saliva efficiently and safely from the mouth to the esophagus and stomach [5]. Oropharyngeal Dysphagia(OD) is prevalent in the elderly and people with complex medical conditions; for example, aspiration pneumonia is caused by silent aspiration in stroke patients, resulting in considerable medical psychosocial consequences, reduced quality of life, and affected prognosis of patients. Prevalence estimates for OD determined by a meta-analysis were 36.5% in hospital settings, 42.5% in rehabilitation settings, and 50.2% in nursing homes [6].

The type and combination of screening and assessment methods used to determine the prevalence of OD varies. More than half of the studies reported OD prevalence data using screening and clinical non-instrumental assessment methods or tools that were either designed by the authors for the purpose of the study or modified versions of published tools. Thus lacking information on diagnostic performance and psychometric properties [7–9]. However, instrumental assessments require specialized training and equipment because of feasibility (e.g., availability, ease of administration). Screening and clinical non-instrumental assessments are the natural first choice for estimating the prevalence of OD [10]. Speech-language pathologists and other specialists, in collaboration with family physicians, can provide structured assessments and make appropriate recommendations for safe swallowing [11]. One study's results suggested that continuing education hours were significantly associated with dysphagia screening protocol choice [12]. One study evaluated nurses' barriers to compliance with dysphagia measures. And it demonstrated that a knowledge deficit was an important barrier in dysphagia care for nurses, and this can be improved with short training [13].

Studies on the factors contributing to barriers to the assessment of dysphagia are lacking. Some Chinese researchers investigated the practice and management of medical staff's dysphagia assessment, it showed that More than 96.0% of the nurses did not know anything about the swallowing function training, and only 36.7% of the nurses used the scale to screen the patients for swallowing disorders [14]. And these studies did not explore the influencing factors. The

study aimed to explore factors contributing to barriers to the assessment of dysphagia, to provide reference suggestions for constructing the training program and improving the rate of dysphagia assessment by conducting this survey.

## Methods

### Design and participants

This was a cross-sectional study. A total of 430 nurses and doctors were selected by convenience sampling from four provinces (Guangdong, Hunan, Guangxi, and Shaanxi). The inclusion criteria were as follows: (a) Chinese nurse practitioners, doctors, or therapists with a medical background, (b) working at the hospital for more than 3 months, and (c) not internship.

According to the sample content estimation method proposed by Kendall, the sample content of questionnaire survey should be 5–10 times the number of items [15]. There were 46 items in the questionnaire. Thus, 230–460 participants were calculated according to the ratio. Adding a 15.0% attrition rate, the total participants sample should between 265 and 529. A total of 353 nurses and doctors completed the survey in this study.

An anonymous survey was conducted by Longgang central hospital of Shenzhen on May 23–31, 2022. Participants were informed about the aims and content of the study and the importance of enrollment. A total of 430 nurses and doctors completed the questionnaire, corresponding to a response rate of 100.0%. The valid questionnaire was 82.1%, and invalid questionnaires were eliminated. The criteria for eliminating invalid questionnaires were (a) incomplete questionnaires, (b) a questionnaire with incorrect content, (c) logically contradictory questionnaires, and (d) time to fill in the questionnaire is shorter than 300 seconds and longer than 3600 seconds.

### Ethical statements

This study was approved by Ethics Committee of Longgang Central Hospital of Shenzhen (No. 2022ECPJ101). The first part of the questionnaire mainly included informed consent, participants will read the informed consent at first, if they agree to participate in the study they will chose "I agree" to complete the following survey. The questionnaire survey was anonymous and personal information was not disclosed.

### Research instruments

The following instrument covered two parts:.

1. **The general information sheet.** Demographic characteristics were measured using the general information sheet, which includes 11 items: province, hospital level and type, department, position, title, working years in the field of dysphagia-related diseases, education, experience of nursing patients with dysphagia, related training for dysphagia, specialized training (geriatric, swallowing and rehabilitation).

2. **Questionnaire on medical staff's knowledge, attitudes and behavior of dysphagia assessment.** This questionnaire was revised based on the existing questionnaires designed by Dr. Xiaofang Dong [16] and Master Keke Ma [17] from The First Affiliated Hospital of Zhengzhou University. Before the study, we obtained the consent of the author of the original questionnaire to add the scoring method and paraphrase the items of knowledge dimension on the basis of the original questionnaire. The K, A and B was shortened for Knowledge, Attitudes and Behavior respectively. The MS-KAB-DA was shortened for the Questionnaire

of Medical staff's Knowledge, Attitudes, and Behavior of dysphagia assessment. The questionnaire was composed of three specific domains, namely Knowledge (25 items), Attitude (8 items), and Behavior (13 items). The knowledge domain consisted of 17 True or False questions (KR-20 = 0.760), four single-choice questions, and four multiple-choice questions. The scoring methods and rules of the questionnaire were shown in S1 Table. The total full score of Knowledge, Attitude and Behavior were 24.4, 40 and 65, respectively. Higher scores indicated higher cognitive, willingness and practice levels. Internal consistency estimates were shown to have acceptable reliability for the questionnaire before the survey (Cronbach's alpha for domain K, A, and B are 0.77, 0.97 and 0.91, respectively).

The instrument was administered in Chinese language and designed as a google form, the website was https://www.wjx.cn/vm/mc5kOo1.aspx.

### Data collection and analysis

All participants were invited to complete the google form through WeChat, DingTalk, and Tencent instant messenger. A professional questionnaire survey platform which provides functions equivalent to Amazon Mechanical Turk called "Wenjuan Xing" was used to investigate. The researcher sent the google form to colleagues and classmates to fill in and asked them to forward to their colleagues. Data were analyzed using SPSS for Windows, version 23. Statistical significance was set at $p < 0.05$. The findings were summarized using descriptive statistics, univariate analysis, and dummy multiple regression analysis. The data were normally distributed by skewness and kurtosis tests. Descriptive statistics (e.g., mean ± SD, frequency & %) were used to summarize the study variables. Univariate analysis (independent-samples t-test and one-way analysis of variance for categorical independent variables) was performed to explore the potential predictors of medical staff's K, A and B for dysphagia assessment. One-way analysis of variance and post hoc multiple comparisons were conducted to analyze the differences in medical staff's K, A and B for dysphagia assessment among four Provinces.

## Results

### Characteristics of the participants

A total of 430 nurses and doctors were enrolled, of which 353 nurses and doctors were valid. Clinical nurses, clinicians, and managers accounted for 66.3%, 19.3%, and 9.6%, respectively. Of the included hospitals, 81.0% were tertiary hospitals. Other characteristics, such as department, medical staff work, and learning experience are summarized in Table 1.

### K, A and B scores

K, A and B scores and their standardized scores by percentage were listed in Table 2. The knowledge and behavior scores were medium, but the attitude scores were high.

### K, A and B differences in 4 provinces

As seen in Table 3, medical staff in Shaanxi province were more willing to assess dysphagia and assessed more frequently than other provinces.

### Univariate analysis

K scores were compared among the different demographic subgroups. Detailed findings were presented in Table 4. The knowledge of medical staff was not significantly different between hospital type and positions.

**Table 1. Demographic characteristics of the study participants (n = 353).**

| Participants'characteristics | | Guangdong n (%) | Hunan n (%) | Shaanxi n (%) | Guangxi n (%) |
|---|---|---|---|---|---|
| **Hospital level** | Level 1 | 13 (6.2) | 1 (1.3) | 0 (0.0) | 0 (0.0) |
| | Level 2 | 49 (23.2) | 3 (3.4) | 0 (0.0) | 1 (4.0) |
| | Level 3 | 149 (70.6) | 73 (95.3) | 40 (100.0) | 24 (96.0) |
| **Hospital type** | The general hospital | 194 (91.9) | 77 (100.0) | 40 (100.0) | 25(100.0) |
| | Other hospital | 17 (8.1) | 0 (0.0) | 0 (0.0) | 0 (0.0) |
| **Department** | Department (Neurology, Rehabilitation, Geriatrics) | 80 (37.9) | 39 (50.6) | 38 (95.0) | 20 (80.0) |
| | Other department | 131 (62.1) | 38 (49.4) | 2 (5.0) | 5 (20.0) |
| **Position** | Clinical nurse | 131 (62.1) | 51 (66.2) | 38 (95.0) | 14 (56.0) |
| | Clinical doctor | 35 (16.6) | 23 (29.9) | 0 (0.0) | 10 (40.0) |
| | Management personnel | 30 (14.2) | 2 (2.6) | 1 (2.5) | 1 (4.0) |
| | Community nurses* | 12 (5.7) | 0 (0.0) | 0 (0.0) | 0 (0.0) |
| | Others | 3 (1.4) | 1 (1.3) | 1 (2.5) | 0 (0.0) |
| **Title** | Primary title | 94 (44.6) | 36 (46.8) | 23 (57.5) | 12 (48.0) |
| | Medium-grade professional title | 91 (43.1) | 30 (38.9) | 17 (42.5) | 10 (40.0) |
| | Senior title of professional | 26 (12.3) | 11 (14.3) | 0 (0.0) | 3 (12.0) |
| **Working years in the field of dysphagia related diseases** | None | 87 (41.2) | 13 (16.9) | 7 (17.5) | 6 (24.0) |
| | <3 years | 32 (15.2) | 24 (31.1) | 5 (12.5) | 2 (8.0) |
| | 3–5 years | 28 (13.3) | 12 (15.6) | 8 (20.0) | 6 (24.0) |
| | ≥5 years | 64 (30.3) | 28 (36.4) | 20 (50.0) | 11 (44.0) |
| **Education** | Junior college and below | 43 (20.4) | 18 (23.4) | 0 (0.0) | 1 (4.0) |
| | Bachelor | 145 (68.7) | 53 (68.8) | 39 (97.5) | 16 (64.0) |
| | Master degree or above | 23 (10.9) | 6 (7.8) | 1 (2.5) | 8 (32.0) |
| **Experience in nursing patients with dysphagia** | Yes | 137 (64.9) | 44 (57.1) | 35 (87.5) | 18 (72.0) |
| | No | 74 (35.1) | 33 (42.9) | 5 (12.5) | 7 (28.0) |
| **Related training for dysphagia** | Yes | 109 (51.7) | 28 (36.4) | 29 (72.5) | 9 (36.0) |
| | No | 102 (48.3) | 49 (63.6) | 11 (27.5) | 16 (64.0) |
| **Specialized training (geriatric, swallowing and rehabilitation)** | Yes | 31 (14.7) | 6 (7.8) | 14 (35.0) | 4 (16.0) |
| | No | 180 (85.3) | 71 (92.2) | 26 (65.0) | 21 (84.0) |

Note.

*Community nurses refer to nursing professionals who are engaged in community nursing work in community health institutions and other relevant medical institutions. Community nursing focuses on families, communities and related groups.

## Predictors of medical staff's K, A and B of dysphagia assessment

In the final regression analysis, all the categorical variables were transformed into dummy variables showed in S4 Table. Dummy multiple regression analysis was conducted to identify relevant predictors for medical staff's K, A and B in dysphagia assessment.

**Table 2. Participants' scores of knowledge, attitudes and behavior in dysphagia assessment.**

| Instruments and score range | Mean ± SD | Percentige rating score | Minimum | Maximum | Range |
|---|---|---|---|---|---|
| **Medical staff's Knowledge of dysphagia assessment (0–24.4)** | 15.3±2.7 | 62.9 | 3.6 | 22.2 | 18.6 |
| **Medical staff's Attitude of dysphagia assessment(8–40)** | 35.9±4.9 | 89.7 | 8.0 | 40.0 | 32.0 |
| **Medical staff's Behavior of dysphagia assessment(13–65)** | 41.4±14.4 | 63.7 | 13.0 | 65.0 | 52.0 |

**Table 3. Differences of medical staff 's knowledge, attitudes and behavior of dysphagia assessment in different province (one-way analysis of variance).**

| Instruments and subdomains | Guangdong | Hunan | Guangxi | Shaanxi | F | p values |
|---|---|---|---|---|---|---|
| | Mean ± SD | | | | | |
| Medical staff's Knowledge of dysphagia assessment(0–24.4) | 15.4±2.5 | 14.9±2.4 | 16.6±2.5 | 15.2±3.7 | 2.519 | 0.058 |
| Medical staff's Attitude of dysphagia assessment(8–40) | 35.7±5.1 | 35.2±4.7 | 37.1±4.1 | 37.7±4.0 | 3.023 | 0.030* |
| Medical staff's Behavior of dysphagia assessment(13–65) | 40.8±15.5 | 38.5±12.0 | 43.5±13.1 | 49.1±11.1 | 5.273 | 0.001** |

Note. Results of post hoc multiple comparisons between provinces

*Medical staff in Shaanxi province were more willing to assess dysphagia than Guangdong(p = 0.037) and Hunan(p = 0.023)

**Medical staff in Shaanxi province were more frequently evaluated for dysphagia than Guangdong(p = 0.001) and Hunan(p<0.001)

As seen in Table 5, Working years in the field of dysphagia related diseases, Specialized training (geriatric, swallowing and rehabilitation); Experience in nursing patients with dysphagia; Department (Neurology, Rehabilitation, Geriatrics) and Related training for dysphagia were significant predictors of medical staff's knowledge of dysphagia assessment, accounting for 31.5% of variance.

As seen in S5 Table, working years in the field of dysphagia related diseases, Department (Neurology, Rehabilitation, Geriatrics) and Title were significant predictors of medical staff's knowledge of dysphagia assessment, accounting for 7.8% of variance.

As seen in S6 Table, Education, Experience in nursing patients with dysphagia, Department (Neurology, Rehabilitation, Geriatrics) and Related training for dysphagia were significant predictors of medical staff's attitude of dysphagia assessment, accounting for 12.9% of variance.

## Discussion

### Status of K, A and B of dysphagia assessment

The knowledge of medical staff on dysphagia assessment directly affects the quality of care and the prognosis of patients. However, most nurses lack the knowledge of dysphagia assessment, which may be related to lack of relevant training and attention paid by medical institutions to the knowledge education of nurses [18]. To build an appropriate and effective training program, it is necessary to combine the influencing factors of nurses' knowledge, attitude and behavior in the assessment of swallowing disorders. There is no research on the influencing factors of knowledge, attitude and behavior of dysphagia assessment in China. This study was conducted in multiple regions (4 provinces), multiple populations (doctors, clinical and community nurses, technicians, etc.) and multiple departments (neurosurgery and general departments) aiming to find as different data and features as possible to serving the training program.

In the study, the K, A and B score were (15.3±2.7), (35.9±4.9) and (41.4±14.4) respectively. According to the percentile rating score in Table 2, Knowledge and Behavior scores were medium, and attitude scores were high. This showed that the attitude score of medical staff is higher than the behavior score because the standardized score of attitude is 89.7, which is much higher than the standardized score of behavior 63.7.

The results of this study are different from study of Sun Qian et al. [19]. The percentige rating score of knowledge and attitude in this study is similar to study of Sun Qian, which is at the medium and high level respectively. However, the percentige rating score of behavior in this study is slightly lower than study of Sun Qian (63.7 VS 71.0). The reasons may be as follows: Firstly, different participants and locations were surveyed in two studies. Sun Qian

**Table 4. Differences of medical staff 's knowledge of dysphagia assessment in sociodemographic, training and working experience characteristics (n = 353).**

| Characteristic | | n(%) | K Mean ± SD | 95%CI | Univariate analysis (t/F, p) |
|---|---|---|---|---|---|
| **Hospital level** | Level 1 | 14 (4.0) | 13.7±1.9 | (12.6,14.8) | F = 3.082, p = 0.047 |
| | Level 2 | 53 (15.0) | 15.1±2.4 | (14.5,15.8) | |
| | Level 3 | 286 (81.0) | 15.5±2.7 | (15.1,15.8) | |
| **Hospital type** | The general hospital | 336 (95.2) | 15.4±2.7 | (15.1,15.7) | t = 0.862, p = 0.389 |
| | Other hospital | 17 (4.8) | 14.8±2.7 | (13.4,16.2) | |
| **Department** | Department (Neurology, Rehabilitation, Geriatrics) | 177 (50.1) | 15.9±2.8 | (15.5,16.3) | t = 3.798, p<0.001 |
| | Other department | 176 (49.9) | 14.8±2.4 | (14.5,15.2) | |
| **Position** | Clinical nurse | 234 (66.3) | 15.2±2.8 | (14.8,15.6) | F = 1.242, p = 0.293 |
| | Clinical doctor | 68 (19.3) | 15.6±1.9 | (15.1,16.1) | |
| | Management personnel | 34 (9.6) | 16.1±2.9 | (15.1,17.1) | |
| | Community nurses | 12 (3.4) | 14.7±3.0 | (12.8,16.6) | |
| | Others | 5 (1.4) | 15.3±1.5 | (13.4,17.1) | |
| **Title** | Primary title | 165 (46.7) | 14.9±2.5 | (14.5,15.3) | F = 8.475, p<0.001 |
| | Medium-grade professional title | 148 (41.9) | 15.5±2.8 | (15.0,16.0) | |
| | Senior title of professional | 40 (11.3) | 16.7±2.1 | (16.0,17.4) | |
| **Working years in the field of dysphagia related diseases** | None | 113 (32.0) | 14.6±2.7 | (14.1,15.1) | F = 7.643, p<0.001 |
| | <3 years | 63 (17.9) | 15.2±2.3 | (14.7,15.8) | |
| | 3–5 years | 54 (15.3) | 15.2±2.7 | (14.4,15.9) | |
| | ≥5 years | 123 (34.8) | 16.2±2.6 | (15.7,16.6) | |
| **Education** | Junior college and below | 62 (17.5) | 14.5±2.4 | (13.9,15.1) | F = 4.267, p = 0.015 |
| | Bachelor | 253 (71.7) | 15.5±2.8 | (15.1,15.8) | |
| | Master degree or above | 38 (10.8) | 15.8±1.8 | (15.2,16.4) | |
| **Experience in nursing patients with dysphagia** | Yes | 234 (66.3) | 15.7±2.8 | (15.3,16.0) | t = 3.287, p = 0.001 |
| | No | 119 (33.7) | 14.7±2.3 | (14.3,15.1) | |
| **Related training for dysphagia** | Yes | 175 (49.6) | 15.8±2.9 | (15.4,16.2) | t = 3.279, p = 0.001 |
| | No | 178 (50.4) | 14.9±2.4 | (14.5,15.2) | |
| **Specialized training (geriatric, swallowing and rehabilitation)** | Yes | 55 (15.6) | 16.3±3.3 | (15.4,17.2) | t = 3.011, p = 0.003 |
| | No | 298 (84.4) | 15.2±2.5 | (14.9,15.4) | |

Note.Abbreviation: CI = confidence interval, K = Knowledge.

A scores were compared among the different demographic subgroups. Detailed findings were showed in S2 Table. The attitude of medical staff was not significantly different between hospital level and title.

B scores were compared among the different demographic subgroups. Detailed findings were showed in S3 Table. The behavior of medical staff was not significantly different between position, title and education.

**Table 5. Factors related to behavior of medical staff by stepwise regression.**

| Variables | Std. β | t | p value |
|---|---|---|---|
| Working years in the field of dysphagia related diseases (ref: <3 years): none | -0.105 | -2.003 | 0.046 |
| Working years in the field of dysphagia related diseases (ref: <3 years): 3–5 years | 0.035 | 0.759 | 0.449 |
| Working years in the field of dysphagia related diseases (ref: <3 years): ≥5 years | 0.014 | 0.280 | 0.780 |
| Specialized training (geriatric, swallowing and rehabilitation) (ref: No): Yes | 0.134 | 2.785 | 0.006 |
| Experience in nursing patients with dysphagia(ref: No): Yes | 0.164 | 3.235 | 0.001 |
| Department (Neurology, Rehabilitation, Geriatrics) (ref: No): Yes | 0.249 | 4.691 | <0.001 |
| Related training for dysphagia(ref: No): Yes | 0.183 | 3.656 | <0.001 |

note. Std. β, standardized βcoefficient

Adjusted $R^2$ = 0.315, F = 33.349, p<0.001

surveyed nurses in 17 Grade III hospitals in Beijing, while this study surveyed medical staff in hospitals of different levels in four provinces. Secondly, different departments were surveyed in two studies. Sun Qian only investigated the department of neurosurgery, which is known to pay more attention to the evaluation practice of swallowing disorders than other departments.

The lack of knowledge in this study was reflected in the following aspects: First, 80.4% and 76.3% participants did not know how to perform the volume-viscosity swallowing test (VVST) and water swallow test, respectively. This is related to the fact that 84.4% of the participants did not have Specialized training (geriatric, swallowing and rehabilitation). Only half of the participants came from Neurology, Rehabilitation and Geriatrics department. Second, nearly half of the participants did not know the concept of mouthful size, food requirements for patients with dysphagia, or how to feed patients with hemiplegia. The deficiencies also included using the wrong utensils, incorrect positioning of the patient during feeding, or giving patients fluids or food of the wrong consistency. There's a study demonstrated that Speech-language pathologists varied in their views on the extent of their role in managing mealtime difficulties. Additionally, their self-rated knowledge of mealtime difficulties was lower than their dysphagia knowledge [20]. These indicated that the clinical attention to managing mealtime difficulties with dysphagia is insufficient. A study demonstrated that knowledge deficit was an important barrier in dysphagia care for nurses, and this can be improved with a short training [13]. Further research to develop efficient and effective training for care staff supporting mealtime difficulties and dysphagia is essential.

The lack of behavior in the study was mainly reflected in the following aspects. First, in 79.1% of the participants' departments, when stroke patients were screened for dysphagia, no further instrumental examinations were performed. Second, in 13.9% of the participants' departments, swallowing function was not screened in stroke patients after admission. Nurse stress with patient characteristics and workload may affect whether a swallow screen is undertaken [21]. The time of screening or assessment of OD prevalence was recorded. OD prevalence estimates from the hospital setting were reported as the time post-stroke or time from admission. Time post-stroke ranged from the hyperacute phase [22]; ≤24h post-stroke, to the acute phase [23]; 1–7 days, to the early subacute phase [24]; 7 days-3 months, to the recovery phase [25]. Therefore, we should screen for or assess dysphagia at the right time based on the patient's condition and identify reliable and standardized assessment methods for screening post-stroke dysphagia(PSD) in acute stroke patients to ensure reliable estimates [26].

One study reported that 35.8% of professionals did not know the definition of dysphagia as a swallowing disorder [27]. Lack of knowledge among healthcare professionals can lead to

inappropriate practices and increase the complications of dysphagia, such as aspiration pneumonia and malnutrition. This is an important barrier in the management of patients with dysphagia. One study investigated the Knowledge, Attitude and Practice of healthcare providers in Iran, and the results showed that very few participants were familiar with a standard test for screening and assessment of dysphagia (11.9%). A total of 74.7% were willing to participate in a workshop on dysphagia; the main pitfalls in their country lie in practice [28]. Another study reported that there seems to be limited awareness among ICU practitioners that patients are at risk of dysphagia, particularly as ventilation persists, protocols, routine assessment, and instrumental assessments are generally not used [29]. Therefore, it is important to improve the practice of dysphagia assessment.

## Comparison of medical staff's K, A and B in different provinces

In the study, the A and B of participants in Shaanxi Province was higher than that of participants in other provinces. The reasons for this are as follows. First, 95.0% of the participants in Shaanxi Province were from neurological, rehabilitation, and elderly related departments that mainly treat patients with stroke. There are many situations in the workplace that require the assessment of swallowing disorders. Second, 87.5% of participants in Shaanxi Province had experience in nursing patients with dysphagia.

## Developing training resources by predictors of K, A and B of dysphagia assessment

There is still plenty of room to improve medical staff's Knowledge, Attitudes and Behavior regarding dysphagia assessment. Multiple linear regression results indicated that experience in nursing patients with dysphagia, related training for dysphagia, working years in the field of dysphagia related diseases, specialized training (geriatric, swallowing and rehabilitation) and department (Neurology, Rehabilitation, Geriatrics) were significant predictors of Behavior. Consequently, the following countermeasures were proposed. The following countermeasures were based on the theory of Knowledge-Attitude-Practice and Self-efficacy.

## Strengthening relevant training to improve knowledge of dysphagia assessment

Knowledge-attitude-practice (KAP) theory is widely used in behavioral psychology [30]. This theory emphasizes that change in an individual's behavior consists of three continuous processes: acquiring knowledge, establishing beliefs and producing behavior, and paying attention to their causality and progressive relationship. The diagnosis and management of dysphagia requires comprehensive knowledge of diverse etiologies, with a systematic approach for the assessment of symptoms, selection of investigations, and appropriate treatment to relieve symptoms [31]. Meanwhile, staff who provide mealtime assistance to people with dysphagia require adequate training to ensure that mealtimes are safe and enjoyable [32]. In this study, knowledge consisted of an assessment approach, dysphagia symptoms, mealtime assistance and so on. The mean score of medical staff was 15.3±2.7, which was at the middle level, and 49.6% participants had received related training for dysphagia(35.8% in other departments VS 63.3% in Neurology, Rehabilitation or Geriatrics department). Only 15.6% of the participants had received Specialized training (geriatric, swallowing and rehabilitation) (8.5% in other departments VS 22.6% in Neurology, Rehabilitation or Geriatrics department). Medical staff in Neurology, Rehabilitation or Geriatrics department seldom receive related training for dysphagia, which leads to less knowledge of assessment or screening for dysphagia. Therefore, it is

more difficult for medical staff to establish positive beliefs regarding dysphagia assessment and screening. To standardize the management of dysphagia, hospitals should take the rate of dysphagia assessment as a quality control index and promote training in dysphagia assessment. Examples include the Volume-Viscosity Swallow Test(V-VST), water swallow test, eating assessment Tool-10(EAT-10), Mann Assessment of Swallowing Ability (MASA) and so on.

### Establishing positive belief in assessment of dysphagia

According to KAP theory, beliefs and attitudes are the motive forces for individuals to produce related behaviors [30]. As the results of study, the attitude score of medical staff was 35.9±4.9, which was at the high level. This finding suggested that many people are willing to screen and evaluate patients with dysphagia. This was a good trend. Nurses gained a sense of value by assessing and caring for patients with dysphagia. Dysphagia screening is often the focus of hospitalized stroke patients; however, dysphagia can also occur in other hospitalized patients and outpatients. Dysphagia can be overlooked by nurses and clinicians, and it is important to educate nurses on the importance of dysphagia screening [33].

### Accumulate dysphagia assessment experience through work and study

Self-efficacy refers to an individual's conviction of their capacity to perform a specific activity. Individuals gain self-efficacy over time as they acquire a range of talents, such as social, cognitive, physical, and linguistic abilities, via life experiences [34]. Efficacy expectations are dynamic and are both appraised and enhanced by four mechanisms [35, 36]: (1) enactive mastery experience or successful performance of the activity of interest; (2) verbal persuasion or encouragement, given by a credible source that the individual is capable of performing the activity of interest; (3) vicarious experience or seeing like individuals perform a specific activity; and (4) physiological and affective states such as pain, fatigue, anxiety, hunger, or dizziness associated with a given activity. These mechanisms to drive efficacy expectations are the concepts upon which the intervention is built to encourage behavioral change. In this study, experience in nursing patients with dysphagia, working years in the field of dysphagia related diseases, and department (Neurology, Rehabilitation Geriatrics) were significant predictors of medical staff's Knowledge, Attitudes and Behavior of dysphagia assessment. The results are consistent with the self-efficacy theory. This suggested that if conditions permit, the medical staff in other departments with no experience of dysphagia patients' nursing should transfer to Neurology, Rehabilitation or Geriatrics department to be trained and accumulate relevant experience. Alternatively, the hospital organizes regular workshops for dysphagia assessment and nurses are required to participate in practice.

### Establish systematic inter-professional collaboration in dysphagia management

A study showed that the main theme "limited professional services" describes how patients received little support from healthcare professionals and had to rely on themselves to adapt to life with dysphagia [37]. The common goal of preventing aspiration and rehabilitating patients' ability to swallow safety is based on dysphagia assessment, using appropriate therapeutic interventions, sharing knowledge, and improving skills among professional groups that consist of nurses, physicians, occupational therapists, and speech-language pathologists(SLPs) [38].

### The strengths and limitations of this study

First, participants in this study were from four provinces in China, showing the differences in knowledge, attitude and behavior of medical staff in different regions in dysphagia assessment. Second, this study explored the influencing factors of medical staff's knowledge, attitude and behavior in tdysphagia assessment, and provided guidance and suggestions for the future organization of relevant training. But, this study only investigated hospital and community nurses, and did not investigate healthcare professionals in other different Settings such as nursing home settings. Future studies should focus on the evaluation of dysphagia and provide training for the assessment of swallowing disorders to medical staff in nursing home settings, social health service centers, and palliative care facilities.

## Conclusions

The attitude of Chinese medical staff to dysphagia assessment is positive, but the knowledge and behavior needs to be improved, and the attitude is separated from the knowledge and the behavior. This study findings also implied that nursing experience, training, and work for patients with swallowing disorders could have positive effects on the Knowledge, Attitudes and Behavior of medical staff regarding dysphagia assessment. The medical management department should take various forms to train the knowledge and skills related to the evaluation of swallowing disorders. Hospital administrators should provide relevant resources, such as videos of dysphagia assessment, training centers for the assessment of dysphagia, and swallowing specialist nurses. It is important that health policies fully recognize the role of training and support systems in caring for people with dysphagia.

## Supporting information

**S1 Table. The scoring methods and rules of the questionnaire.**
(DOC)

**S2 Table. Differences of medical staff 's attitudes of dysphagia assessment in sociodemographic, training and working experience characteristics (n = 353).**
(DOC)

**S3 Table. Differences of medical staff 's behavior of dysphagia assessment in sociodemographic, training and working experience characteristics (n = 353).**
(DOC)

**S4 Table. Dummy variable settings.**
(DOC)

**S5 Table. Factors related to knowledge of medical staff by stepwise regression.**
(DOC)

**S6 Table. Factors related to attitudes of medical staff by stepwise regression.**
(DOC)

## Acknowledgments

The authors would like to thank all participated medical staff who took the time to complete survey and made this study possible.

## Author Contributions

**Conceptualization:** Juanhui Chen.

**Data curation:** Juanhui Chen.

**Investigation:** Juanhui Chen, Wenqiu Ye, Xingyun Zheng, Wenna Wu, Yuebao Chen, Yin-juan Chen.

**Writing – original draft:** Juanhui Chen.

**Writing – review & editing:** Juanhui Chen.

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
