## [Decision Letter · Decision Letter 0]

19 Feb 2024

PONE-D-23-41544Predictors of medical staff’s Knowledge, Attitudes and Behavior of dysphagia assessment: A Cross-Sectional studyPLOS ONE

Dear Dr. Chen,

Thank you for submitting your manuscript to PLOS ONE. After careful consideration, we feel that it has merit but does not fully meet PLOS ONE’s publication criteria as it currently stands. Therefore, we invite you to submit a revised version of the manuscript that addresses the points raised during the review process.

**ACADEMIC EDITOR: ** Thank you for submitting your article - based on reviews from peer reviewers, please consider revising your article for consideration for publication. 

We look forward to receiving your revised manuscript.

Kind regards,

Suzanne Rose

Academic Editor

PLOS ONE

Journal Requirements:

"We would like to thank Shenzhen Longgang District Science and Technology 

Innovation Bureau for providing the Unfunded Medical and Health Technology 

Plan(No. LGKCYLWS2020128)"

"The author(s) received no specific funding for this work"

**Additional Editor Comments:**

In addition to the below comments, please review the additional files uploaded entitled comments to the authors and reviewer comments. 

Reviewers' comments:

Reviewer's Responses to Questions

**Comments to the Author**

1. Is the manuscript technically sound, and do the data support the conclusions?

Reviewer #1: Partly

Reviewer #2: Yes

2. Has the statistical analysis been performed appropriately and rigorously? 

Reviewer #1: N/A

Reviewer #2: Yes

3. Have the authors made all data underlying the findings in their manuscript fully available?

Reviewer #1: Yes

Reviewer #2: Yes

4. Is the manuscript presented in an intelligible fashion and written in standard English?

Reviewer #1: No

Reviewer #2: No

5. Review Comments to the Author

Reviewer #1: Thank the editors for inviting me to peer-review this manuscript. The topic is relatively meaningful and exciting. However, there are a lot of issues that need to be corrected before publication. I have some comments and hope that they are helpful to the authors.

Abstract

1. The objective is unclear (too large). In addition, in the Purpose and Introduction, the authors mentioned “stroke”. However, I do not know whether or not “stroke” was a factor associated with the inclusion criteria of participants.

2. The authors should not mention the number of medical staff approached (430 nurses and doctors) in the Methods. The final number of participants should be mentioned (n=353).

Introduction

3. The authors wrote: “Dysphagia occurs in 37–78% of patients after stroke [2-4].”

However, these three references are seemingly not relevant to the sentence above.

4. The Introduction is quite long. I think the authors should divide it into two paragraphs (one for the prevalence of dysphagia and one for studies involving their topic + the objectives).

Methods

5. What does “Chinese RNs” mean?

6. “Based on the suggestion of Hair, Black, Babin, and Anderson[14], a ratio of 10 to 20 participants for each parameter was required for structural equation modeling.”

Please check the reference 14.

7. “Thus, 220–440 participants were followed by a calculation of a 15% attrition rate, which resulted in a total sample of 430 participants in this study.”

This sentence is unclear.

8. “An anonymous survey was conducted with permission from the research ethics committee of the participating hospital.”

This sentence is unclear (the participating hospital).

In addition, please do not repeat the information involving the ethics in the Methods. The authors had an “Ethical statements” part already.

(“Data collection and Analysis

This study was approved by the ethics committee of the study hospital.”)

9. “The inclusion criteria were as follows: (a) Chinese RNs, doctors, or therapists with a medical background, (b) working at the hospital for more than 3 months, and (c) not internship.”

I do not know how the authors identified and approached 430 nurses and doctors. The authors used online platforms to collect data. However, the participants only came from four provinces (Guangdong Province, Hunan Province, Guangxi Province, and Shaanxi Province).

10. “Before the study, I obtained the consent of the author of the original questionnaire to modify the questionnaire. This questionnaire was revised based on the existing questionnaires designed by Dr. Dong Xiaofang[15] and Master Ma Keke[16] from The First Affiliated Hospital of Zhengzhou University.”

First, “I” should be replaced by “we”.

Second, the authors mentioned two references for their questionnaire. However, the reference 15 cannot be identified. Is it a thesis?

The reference 16 cannot be found in Google/PubMed.

Can the authors attach the questionnaire as a supplementary file?

11. “The MS-KAB-DA was shortened for the Questionnaire of Medical staff’s Knowledge, Attitudes, and Behavior of dysphagia assessment, which is composed of three specific domains: Knowledge, Attitude(eight items), and behavior (13 items). The knowledge domain is composed of 25 items, including 17 True or False items (KR-20=0.760), four single-choice items, and four multiple-choice items.”

So, the total questions were 46, right? In addition, what does “KR-20” mean?

“Based on the suggestion of Hair, Black, Babin, and Anderson[14], a ratio of 10 to 20 participants for each parameter was required for structural equation modeling.”

The sample size should be 460 (46*10) to 920 (46*20), right?

Why did the authors mention a sample size of 220 to 440?

12. “Before the study, I obtained the consent of the author of the original questionnaire to modify the questionnaire.

The questionnaire demonstrated good reliability and construct validity; the reported Cronbach's α coefficient was 0.832–0.879, and the content validity was 0.751 [15-16].”

If the authors modified questions, mentioning the reliability and validity of the old questionnaire is meaningless.

13. “Internal consistency estimates were shown to have acceptable reliability for the questionnaire before the survey (Cronbach’s alpha=0.865).”

This data was for the authors’ questionnaire, right? If yes, I think Cronbach’s alpha should be reported separately for each domain (K, A, and B), not all 46 questions.

14. “This study was approved by the ethics committee of the study hospital. Data were collected on May 23–31, 2022. All participants were invited to complete the questionnaire through WeChat, DingTalk, and Tencent QQ from May 23 to May 31, 2022. A professional questionnaire survey platform which provides functions equivalent to Amazon Mechanical Turk called “Wenjuan Xing” was used to investigate. The researcher sent the questionnaire through WeChat, DingTalk, and Tencent QQ to colleagues and classmates to fill in and asked them to forward the questionnaire to their colleagues.”

Do not repeat information involving the period of data collection and online platforms. In addition, was the questionnaire designed as a Google Form?

15. “Univariate analysis (independent-samples t-test and one-way analysis of variance for categorical independent variables) was performed to explore the potential predictors of medical staff KAB for dysphagia assessment.”

Did the authors check the normal distribution of data before using the t-test and ANOVA test?

Results

16. Please do not stick the numbers and the brackets. For example, 13(6.16). It should be “13 (6.16)”.

17. Please rename columns in the first row of Table 1.

18. Can the authors add some sentences to introduce community nurses?

19. In Table 1, the variable “title” is relatively difficult to understand.

20. A third of participants were not working in the field of dysphagia-related diseases. Why did the authors invite them to participate in this study? It is obvious that knowledge involving dysphagia is unnecessary for them.

Why did the authors not focus on selecting healthcare professionals working in different settings with populations at risk for dysphagia?

21. In Table 2, please explain “percentige rating scores” (how to calculate). Mean/maximum score, right?

This value for the behavior score is inappropriate because the minimum score was 13, not 0.

22. In Tables 2, 3, 4, and 6, I do not think adding the knowledge, attitude, and behavior scores into a total score is a good idea. These three domains are different.

23. “As seen in Table 3, medical staff in Shaanxi Province were more willing to assess dysphagia and were evaluated more frequently.”

I do not understand this sentence.

In addition, sentences in the “Note” part are difficult to understand.

24. Table 5 should be removed or moved to the Supplementary Files. The authors can mention reference groups in Table 6.

For example, related training for dysphagia (ref: No): Yes.

Discussion

25. “This suggests that people are willing to assess swallowing disorders, but lack Knowledge and Behavior.”

This sentence is unclear.

26. Please add a paragraph involving the strengths and limitations of this study.

27. Please check the references. I do not know what [J] means.

28. Please check spelling and grammar mistakes.

Best wishes to the authors.

Reviewer #2: This is a very interesting study and was an enjoyable read. Please see specific comments that will add strength and clarity to the study information presented regarding this topic of predictors of medical staff’s Knowledge, Attitudes, and Behavior of dysphagia assessment. The manuscript needs proofreading from an agency specialized in this task.

Comments attached in PDF file & word document.

6. PLOS authors have the option to publish the peer review history of their article (what does this mean?). If published, this will include your full peer review and any attached files.

Reviewer #1: No

Reviewer #2: **Yes: **Associate Prof Mohammad N. Alshloul, Al-Balqa Applied University, Amman-Jordan

---

## [Author Response · Author response to Decision Letter 0]

29 Feb 2024

Thank you for your letter and for the reviewer’s comments concerning our manuscript entitled “Predictors of medical staff’s Knowledge, Attitudes and Behavior of dysphagia assessment: A Cross-Sectional study”(ID:PONE-D-23-41544). Those comments are all valuable and very helpful for revising and improving our paper, as well as the important guiding significance to our researches. We have studied comments carefully and have made correction which we hope meet with approval. Please find a file named response to reviewers in attach file.

---

## [Decision Letter · Decision Letter 1]

17 Mar 2024

PONE-D-23-41544R1Predictors of medical staff’s Knowledge, Attitudes and Behavior of dysphagia assessment: A Cross-Sectional studyPLOS ONE

Dear Dr. Chen,

Thank you for submitting your manuscript to PLOS ONE. After careful consideration, we feel that it has merit but does not fully meet PLOS ONE’s publication criteria as it currently stands. Therefore, we invite you to submit a revised version of the manuscript that addresses the points raised during the review process.

**Thank you for your revisions to the manuscript. We are appreciative of your work to date in revising the manuscript. Please refer to the further comments provided by the reviewer in order to be able to accept the article for publication. **

We look forward to receiving your revised manuscript.

Kind regards,

Suzanne Rose

Academic Editor

PLOS ONE

Reviewers' comments:

Reviewer's Responses to Questions

**Comments to the Author**

1. If the authors have adequately addressed your comments raised in a previous round of review and you feel that this manuscript is now acceptable for publication, you may indicate that here to bypass the “Comments to the Author” section, enter your conflict of interest statement in the “Confidential to Editor” section, and submit your "Accept" recommendation.

Reviewer #1: (No Response)

Reviewer #2: All comments have been addressed

2. Is the manuscript technically sound, and do the data support the conclusions?

Reviewer #1: Partly

Reviewer #2: Yes

3. Has the statistical analysis been performed appropriately and rigorously? 

Reviewer #1: No

Reviewer #2: Yes

4. Have the authors made all data underlying the findings in their manuscript fully available?

Reviewer #1: Yes

Reviewer #2: Yes

5. Is the manuscript presented in an intelligible fashion and written in standard English?

Reviewer #1: Yes

Reviewer #2: Yes

6. Review Comments to the Author

Reviewer #1: Thanks to the authors for answering my questions and comments in detail. I also have some comments and hope that they are useful for the authors.

1. In the Abstract, the authors should mention whether the knowledge, attitude, and behavior of medical staff were good or poor, positive or negative.

2. In the Data analysis, the authors mentioned, “A comparison analysis was conducted to analyze the differences in medical staff’s KAB for dysphagia assessment among four Provinces.”

Tests used should be described in detail.

3. For my comment, “Can the authors add some sentences to introduce community nurses?”, I do not recommend the authors count the number of community nurses in each hospital.

I mean their definitions (at the bottom of Table 1). Because in my country, we do not have “community” nurses (I rarely hear about them). I think we should have definitions for occupations that are not internationally popular (if necessary - authors' choice).

4. I still do not think adding the knowledge, attitude, and behavior scores into a total score is a good idea. These three domains are different.

For example, working experience is a predictor of medical staff’s KAB. It means that working experience is associated with all their knowledge, attitudes, and behavior. Is it true? However, in this study, “their knowledge and behavior scores were medium, but the attitude scores were high”. In addition, the authors did not demonstrate the validity of the questionnaire (EFA and CFA) to analyze the relationship among three domains (K, A, and B). This means combining these three domains into one is meaningless.

Best wishes to the authors.

Best regards.

Reviewer #2: While offering congratulations, here are some tips to enhance the quality of the manuscript. Revise the notes in the PDF file in your main manuscript & ensure everything is changed accordingly.

7. PLOS authors have the option to publish the peer review history of their article (what does this mean?). If published, this will include your full peer review and any attached files.

Reviewer #1: No

Reviewer #2: **Yes: **Dr. Mohammad N. Alshloul/Associated Professor, Al-Balqaʼ Applied University, Prince Al Hussein Bin Abdullah II Academy for Civil Protection, Amman-Jordan

https://orcid.org/0000-0001-9448-9369

---

## [Author Response · Author response to Decision Letter 1]

19 Mar 2024

Thank you for your letter and for the reviewer’s comments concerning our manuscript entitled “Predictors of medical staff’s Knowledge, Attitudes and Behavior of dysphagia assessment: A Cross-Sectional study”(ID:PONE-D-23-41544). Those comments are all valuable and very helpful for revising and improving our paper, as well as the important guiding significance to our researches. We have studied comments carefully and have made correction which we hope meet with approval.

We submited four items as follow:

1.Response to Reviewers 

2.Revised manuscript with track changes

3.Manuscript (revised) 

Below, please find a response to each of the reviewer’s comments. We have revised manuscript with track changes. We hope that the revisions in the manuscript and our accompanying responses will meet editors’ and reviewers’ criteria.

We shall look forward to hearing from you at your earliest convenience.

Yours sincerely

Juanhui Chen

---

## [Editor Report · Decision Letter 2]

21 Mar 2024

Predictors of medical staff’s Knowledge, Attitudes and Behavior of dysphagia assessment: A Cross-Sectional study

PONE-D-23-41544R2

Dear Dr. Chen,

We’re pleased to inform you that your manuscript has been judged scientifically suitable for publication and will be formally accepted for publication once it meets all outstanding technical requirements.

Kind regards,

Suzanne Rose

Academic Editor

PLOS ONE

Additional Editor Comments (optional): Thank you so much for the expedient response to the reviewer's final comments on the manuscript. It has been a privilege supporting this manuscript in the review process. 
---

## [Editor Report · Acceptance letter]

27 Mar 2024

PONE-D-23-41544R2 

PLOS ONE

Dear Dr. Chen, 

I'm pleased to inform you that your manuscript has been deemed suitable for publication in PLOS ONE. Congratulations! Your manuscript is now being handed over to our production team.

Kind regards, 

on behalf of

Dr. Suzanne Rose 

Academic Editor

PLOS ONE